# Selection of Parameters for Optimized WAAM Structures for Civil Engineering Applications

**DOI:** 10.3390/ma16134862

**Published:** 2023-07-06

**Authors:** Saham Sadat Sharifi, Sebastian Fritsche, Christoph Holzinger, Norbert Enzinger

**Affiliations:** 1Institute of Materials Science, Joining and Forming at Graz University of Technology, Kopernikusgasse 24/I, 8010 Graz, Austria; sebastian.fritsche@tugraz.at (S.F.); norbert.enzinger@tugraz.at (N.E.); 2Institute of Structural Design, Graz University of Technology, Technikerstrasse 4/IV, 8010 Graz, Austria; christoph.holzinger@tugraz.at

**Keywords:** CMT, welding strategies, wire arc additive manufacturing, microhardness

## Abstract

Using the CMT (Cold Metal Transfer, F. Fronius, Upper Austria) welding process, wire arc additive manufacturing (WAAM) enables companies to fabricate steel components in a resource-saving manner (additive vs. subtractive) by properly reinforcing existing steel components. Two fundamental questions are discussed in the current work. The first focus is on the general geometric possibilities offered by this process. The influence of various parameters, such as wire feed speed, travel speed, and torch inclination on the seam shape and build-up rate are presented. The microstructure of the manufactured components is evaluated through metallography and hardness testing. Based on the first results, print strategies are developed for different requirements. Moreover, suitable process parameter sets are recommended in terms of energy input per unit length, weld integrity and hardness distribution. The second focus is on testing and determining joint properties by analyzing the microhardness of the welded structures. The chosen parameter sets will be investigated, and steel quality equivalents according to ÖNORM EN ISO 18265 will be defined.

## 1. Introduction

Construction processes are often associated with high resource consumption, associated CO_2_ emissions, and a limited degree of industrialization in terms of the total value added. Highly relevant, globally driven discussions in industry and business about the use of energy and the limitations of resources suggest a focus on the development of new production strategies. At the same time, increasing digitization in a dynamically networking society is leading to a growing desire for individualized products and flexibility of manufacturing methods [1,2].

In recent decades, the essential characteristics of steel construction have changed significantly. Some 50–100 years ago, labor was cheap and materials were expensive; hence, care was taken to use materials very efficiently right from the planning stage. This is particularly evident in the halls and bridges built at the turn of the 20th century [3]. Optimized, elegant structures, which still stand today as examples of outstanding engineering, were created as a result of this attitude [4]. Due to industrial progress, more and more modern and efficient equipment, and last but not least, the supply and exploitation of low-wage countries, building materials can be made available cheaply and in almost arbitrary quantities. Therefore, optimization of a planned structure is often not economically viable since more significant material inputs can reduce planning and, above all, manufacturing costs to a minimum. Aspects such as sustainability and resource efficiency are thus often inadequately addressed.

In order to counteract these developments, which are not limited to the construction industry, additive manufacturing is seen as a promising production method in all industrial sectors.

WAAM is an efficient method for engineering structure production. The process can produce near-net-shaped components without complex tools, and with lower costs and time consumption. In this method, a robotic arm controls the process, and the shape is built upon a substrate material. WAAM is the process of depositing layers by melting metal wire using an electric arc as the heat source to 3D print metal components [5].

As established as the process already is in mechanical engineering, WAAM is not yet used in the construction sector. The architectural sector generally accounts for only 3% of the annual volume produced by additive manufacturing (across materials, including plastics and concrete/mortar). Additive manufacturing by WAAM in the construction sector is not yet included in any statistics [6].

Currently, no WAAM process is used in the construction industry beyond the field of materials and process research. Conventional and proven processes still seem to be the more economical alternative at present.

Current WAAM research projects in civil engineering are the steel bridge welded by MX3D (https://mx3d.com/industries/design/smart-bridge/ (accessed on 9 June 2023)) and projects at RWTH Aachen [7] and TU Darmstadt [8,9,10,11]. As in most application areas of additive manufacturing, the scale of the manufactured structures is also of interest in the WAAM application field. WAAM offers the possibility to produce in scales relevant to civil engineering. This can range from local reinforcements of existing geometries to the production of entire structures. However, the aim of the ongoing 3DWelding project is to develop a resource-saving, sustainable, and economically viable manufacturing method in structural steel engineering that enables leaner and lighter structures while ensuring load-bearing capacity and serviceability.

The results published in this paper together with the results presented in [12] are intended to show that the method is applicable according to current standards in the construction industry. All requirements according to Eurocode 3 and EN 10025-2 [13] are met.

Knowledge of the characteristics and mechanical properties of the weld is of significant importance for an assessment of the properties of AM components. In this study, experimental tests were carried out to investigate the macrostructure, microstructure, and microhardness of the AM build-up of plain carbon steel using the CMT welding process. In the following, the results of the properties of the weld and, subsequently, the basis of the dependence of the microhardness on the energy input are presented. This work forms a significant basis for the adaptation of the welding parameters to the mechanical properties of the materials and serves as a starting point for further investigations.

## 2. Materials and Methods

The welding wire was in the form of industrially processed PC wire rods measuring 1.2 mm in diameter. The filler material used to print the investigated structures was Böhler EMK6, an unalloyed solid wire of type G 3Si1 with a yield strength of 440 N/mm^2^. A shield gas with 10% CO_2_ and 90% Ar was used for all welds.

The chemical composition of the substrate material is given in Table 1. A structural sample was chosen to evaluate the feasibility of a wire arc additively manufactured component. Figure 1 illustrate the schematic of the WAAM process. Figure 2 shows an exemplary part that is manufactured by WAAM. The substrate in this study is plain carbon steel S355JR.

The welds were optically studied with a Zeiss stereo microscope, model Discovery.V20 (Oberkochen, Germany), and the images were processed using ImageJ software. In addition, a Zeiss Observer Z1m microscope (Carl Zeiss Microscopy, Jena, Germany) was used for the microstructural analysis. The samples were ground and polished, and were followed by etching with 3% Nital.

The microhardness test was carried out for selected samples using the EMCO TEST M1C 010 machine. For hardness investigation along the vertical build direction, a cross-section was taken from the samples, and all samples were ground and polished to obtain a mirror surface. Moreover, Vickers hardness (HV1) was measured from the top to the bottom of each section according to the EN ISO 6507 standard [15]. Concerning the penetration depth, the microhardness of each sample was measured up to 5 rows (approx. 1.8 mm) beneath the HAZ. In the end, the maps of microhardness analysis were obtained using OriginLab 2019 software.

For convenient follow-up, we define Walls as single-track and multi-layer welds, and Blocks as multi-track and multi-layer welds. Here, the track is considered to be the number of weld seams on the substrate.

## 3. Parameter Selection

Single-track, single-layer welds were carried out using the CMT process variant. For the standard CMT process, the WFS and TS were varied in the range of 1.5 m/min to 8.5 m/min and 3 mm/s to 25 mm/s, respectively. In addition, the cross-sections were ground, polished, and etched with 3% Nital to reveal the microstructure of the weld and to measure its dimension and different area fractions, which are presented in Figure 3.

Based on the single-track weld observations, suitable parameters are selected for a single-track and five-layer welding investigation. Table 2 represents the selected parameters.

Furthermore, for geometry analysis, samples were welded as Walls and Blocks with different overlaps using parameter set A to F according to Figure 4. In the current work, for the given parameter configuration, the distance between the welds is defined as 60%, 66%, 75%, and 80% (f_a_) of the width of the weld bead, i.e., the b_w_ (Figure 4a). Wall and Block welds were subjected to macro and micro analysis to determine the influence of the individual parameters on the weld geometry, i.e., the gap between tracks, penetration, average width, and average height.

## 4. Results and Discussion

### 4.1. Single-Layer and Single-Track Welds

Figure 5 depicts the connection of the geometry of the weld cross-section to both WFS and TS. The greater values for weld track height are achieved for low TS. The higher the TS, the stronger the influence of the WFS on the track height (Figure 5a). Low WFS leads to a narrower track, and with increasing WFS, the influence of TS is higher, resulting in the widest track for high WFS combined with high TS (Figure 5b). For a sufficiently built WAAM structure, a suitable combination of the width and height of a track is crucial to fulfil the requirements regarding surface quality and efficiency. Therefore, based on the geometrical parameters of the weld cross-section, a non-suitable parameter combination can be excluded from further investigations. Due to high heat input, leading to high penetration depth and dilution, a WFS larger than 4.5 m/min could be determined as insufficient (Figure 5c). The CO_2_ in the shielding gas promotes penetration and leads to higher heat input and, therefore, a wider weld bead [4]. In Figure 5d, the highest cross-sectional area, and therefore the highest deposited volume per length, can be observed for high WFS and TS. This correlates with the calculations by Plangger et al. [16], who calculated A_Q_ with respect to TS and WFS.

Considering the width-to-height ratio, integrity, and the flank angle of the welded track, suitable WFS and TS were observed to be in the range of 1.5 m/min to 4.5 m/min and 3 mm/s to 7.5 mm/s, respectively. Therefore, parameter sets A to F were selected for further investigation (Figure 6). The current and voltage measured during the welding process with different WFS are presented in Table 3. Estimating the heat input without considering losses according to [16] using the ratio of WFS to TS, where C represents the lowest and D the highest heat input, was also observed from the cross-section micrographs. Specimens conducted with B and F showed a similar heat input per unit length.

### 4.2. Wall Welds

Figure 7 shows the superposition of hardness maps and microhardness measurements of the selected parameters.

Microhardness plots for each parameter setup are shown in Figure 8. Each value is an average of all measured values in a row. The baseline is defined as the surface of the base metal (substrate). The hardness values range between 150 HV1 and 250 HV1, with a standard deviation between 12 and 16.5 HV1. The distribution of the hardness values is almost identical for all parameters throughout the entire cross-section, and the local variation seems to be due to the local tempering effects. Furthermore, the hardness in HAZ increases in samples welded with parameters A and C due to the lower heat input. The processes with lower heat input cool faster, leading to a higher hardness, whereas those with high heat input have a slower cooling rate, leading to a lower hardness (Figure 7).

In welding, there exists a relationship between the power (P = I × V) and the wire feed speed. This relationship can vary depending on the specific welding process being used. In CMT welding, the power input is directly related to the wire feed speed. This relationship is often nearly linear, indicating that a higher wire feed speed, which corresponds to a higher deposition rate, results in a higher power input [16]. This relationship can be described by Equation (1):(1)P= κ×WFS
where P represents power, WFS represents wire feed speed, and κ is a constant that represents the proportionality factor.

To correlate the microhardness distribution to welding parameters, we used energy input per unit length, E. The relationship between power input and energy input per unit length is an important aspect to consider for process control and optimization in CMT. Power input refers to the electrical power supplied to the welding system, typically measured in watts (W). Energy input per unit length, on the other hand, represents the total energy consumed in the welding process per unit length of the weld and is commonly measured in joules per millimeter (J/mm). To calculate the energy input per unit length, the power input is divided by the welding speed, taking into account the TS and the WFS [16]:(2)$$E\;=\frac{P}{\mathrm{TS}}=\;\kappa \times \frac{\mathrm{WFS}}{\mathrm{TS}}$$

Figure 9 illustrates the energy input per unit length for each parameter. The results show that the higher the energy input per length, the more homogeneous the microhardness distribution [17] (Figure 10). At higher TS, the weld cools faster. This will reduce the tempering effect and cause relatively higher hardness values, as expected by comparison between samples with minimum and maximum energy inputs, i.e., samples C and D (see Figure 9). However, the parameter with the homogenous hardness distribution, i.e., parameter D, consumes more material, and the weld build-up increases in width and height. On the other hand, although parameter C leads to the lowest energy input, it results in nonhomogenous hardness distributions. Parameter selection is conducted considering weld integrity and wettability, as well as the lowest possible energy input. Therefore, considering the optimum energy input per length, i.e., proportional to 10 kJ/mm, the experimental window was confined to parameters A, B, E, and F. Finally, parameter E was excluded from the selection due to its nonhomogenous hardness distribution.

### 4.3. Single-Layer and Multi-Track Welds

Figure 11, Figure 12 and Figure 13 represent the macrographs of samples welded by parameters A, B, and F. For all overlaps, a lack of fusion is visible in the sample that is welded with parameter A. Welding with parameter B causes a lack of fusion between tracks as the track spacing increases. Parameter F delivers a stable welding shape and a higher penetration depth.

### 4.4. Weld Integrity

Figure 14 shows various geometrical measurements. In the present work, the measured gap is the sum of all the distances between tracks for each overlap. For all overlaps, parameter A results in a large gap between tracks. Parameter B shows a gap between the tracks of magnitude 0.1 mm to 1.4 mm as the overlap increases, and no gap between tracks is detected in the specimen of parameter F. Parameter F shows a high dilution of 1.7 mm to 2.2 mm, while the welds with parameter A almost do not penetrate into the base metal. This indicates that the wire feed speed plays a significant role in welding integrity; as the filler wire goes faster, the power becomes greater and a larger dilution is achieved [1,17].

#### 4.4.1. Effect of Wire Feed Speed on Weld Bead Area

The wire feed speed is a significant factor in the weld bead area. The weld bead cross-section area is a function of the width, height, and depth [3]. Ramos et al. showed that the wire feed speed is correlated to the height and the depth, but its correlation was not statistically significant with the width [2]. Plangger reported that the cross-section area is related directly to the WFS and inversely to the TS [16]. They suggest that high values of the wire feed speed followed by a lower travel speed are advised in order to optimize this region. The welding wire serves as filler material; higher speeds provide more welding material per second [16,18].

#### 4.4.2. Effect of Travel Speed on Weld Bead Area

The travel speed as well as the wire feed speed are significant parameters in optimizing the weld bead area. Travel speed, generally, has a reverse relation to the weld area since welding at a slower pace allows for more filler per unit area [19].

### 4.5. Block Welds

The same parameters are selected for investigating the effect of welding parameters on the geometry and shape of welding beads in multi-layer and multi-track welding. Figure 15, Figure 16 and Figure 17 represent the cross-sectional macrographs of the specimens. Parameter A shows irregular weld shapes for all overlaps. The welds include a lack of fusion regions, which are indicated by arrows (see Figure 15). The parameter B likewise contains defects that are evenly distributed among the layers (indicated in Figure 16). With the lower overlap, i.e., 60%, the lack of fusion region is restricted to the primary layers, while as the overlap increases, these regions tend to appear in the whole weld bead. In other words, with parameter B, as the overlap increases, the size and number of the lack of fusion regions increase. Moreover, specimens that were welded with parameter F did not show defects at lower overlaps (see Figure 17, 60% and 66%). These results help to correlate the weld integrity concerning the welding parameters and structure. Although all the selected parameters had approximately identical energy input per unit length, they lead to different welding integrity. This knowledge could assist in building up a structure with efficient energy consumption and higher integrity.

### 4.6. Block Welds with Combined Parameters

According to the results discussed in previous sections and the input energy per unit length (see Figure 9), to acquire an intact build-up on the one hand and to optimize the energy input and weld geometry on the other, a new welding strategy was set up. In the new welding model, two parameters, including the one that results in an intact weld with lower energy input per length (parameter F), and the one that results in optimized geometry (parameter B), were selected. The heat flow differs in the first layer because the substrate is a large heat sink.

To optimize welding with parameter B with energy consumption and acquire an intact weld (free of lack of fusion regions), two lower overlaps with these particular characteristics (60% and 66%) were welded as follows:-Strategy 1: first layer with parameter F, and second to tenth layers with parameter B.-Strategy 2: first three layers with parameter F, and fourth to tenth layers with parameter B.

Figure 18 illustrates the weld building up.

Figure 19 displays the macroscopic images of the specimen according to Strategy 1. Both overlaps did not show a lack of fusion regions at this magnification. Specimens were studied using light optical microscopy for closer investigation. Figure 20 and Figure 21 show microscopic images of the two representatives of the bottom and top of the weld bead. The microstructure of the samples consists of ferrite grains (white regions) and pearlite islands (black regions) throughout the entire cross-section. In the lower part of the weld, ferrite grains are homogeneously equiaxed because of the tempering effect. In the upper region, acicular ferrite was observed due to faster cooling. The samples were welded with 60% overlap and did not represent a lack of fusion regions in either the lower layers or the top layers of the weld bead (Figure 20). The top region of the 66% overlap did not contain a lack of fusion regions. However, a few defects with a size of 0.03–0.05 mm^2^ were observed in the lower layers (Figure 21).

Figure 22 depicts the macroscopic images of samples welded according to strategy 2. No lack of fusion regions was detected at the macroscale. As before, the samples were subjected to optical microscopy investigation. Figure 23 and Figure 24 represent the microscopic images of these two samples at the lower layers and top of the weld bead. Similarly, the microstructure of the weld consists of ferrite and a small region of pearlite. The entire weld beads of both 60% and 66% overlap contain no lack of fusion regions.

## 5. Conclusions

The present work reports the influence of the processing parameters of the wire arc additive manufacturing (WAAM) technique on the microhardness and geometry of the weld. The results reveal that wire feed speed and travel speed impact the hardness and distribution throughout the weld bead. Wire feed speed and travel speed are correlated to energy input. Higher heat input leads to a homogeneous hardness distribution.

Experiments also showed that process parameters, e.g., travel speed, wire feed speed, and subsequently the energy input per unit length, influence the bead area in the WAAM process. Two strategies were selected to optimize welding energy as well as wire consumption, and to acquire an intact weld (free of lack of fusion regions). The microscopic investigations of welding according to strategy 1 demonstrated defects in the preliminary layers with 66% overlap, while samples with 60% overlap revealed no defects. Strategy 2 did not contain defects throughout the whole cross-section area of the welds for both 60% and 66% overlaps. Finally, welding the first three layers with a WFS of 4.5 m/min and TS of 7.5 mm/s, and continuing with WFS of 2.5 m/min and TS of 4.0 mm/s was selected as the optimal welding parameter configuration.

It is beneficial to select different parameters for the bottom layers and the rest of the structure to obtain proper integrity and surface roughness. This helps to face the variation in heat flow towards the substrate (first three layers) and built structure.

## Figures and Tables

**Figure 1 materials-16-04862-f001:**
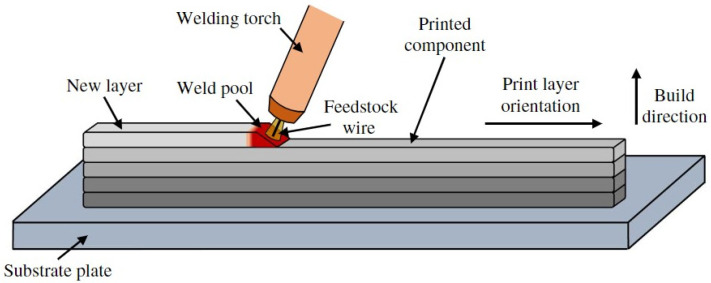
Schematic of the WAAM process [14].

**Figure 2 materials-16-04862-f002:**
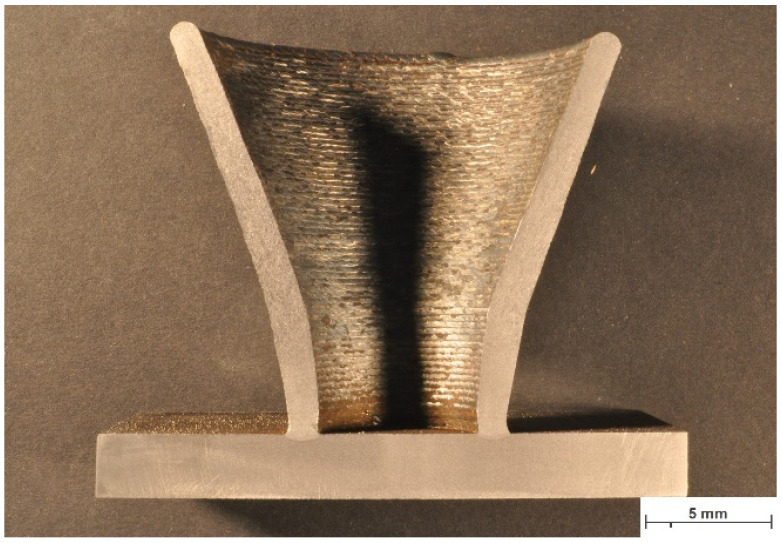
Macroscopic view on a cross-section of an exemplary part manufactured by WAAM.

**Figure 3 materials-16-04862-f003:**
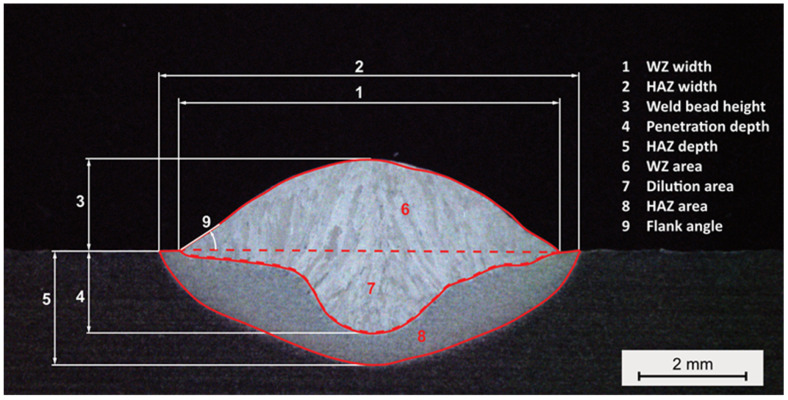
Measurements taken for Analysis of geometrical features of the weld cross-section.

**Figure 4 materials-16-04862-f004:**
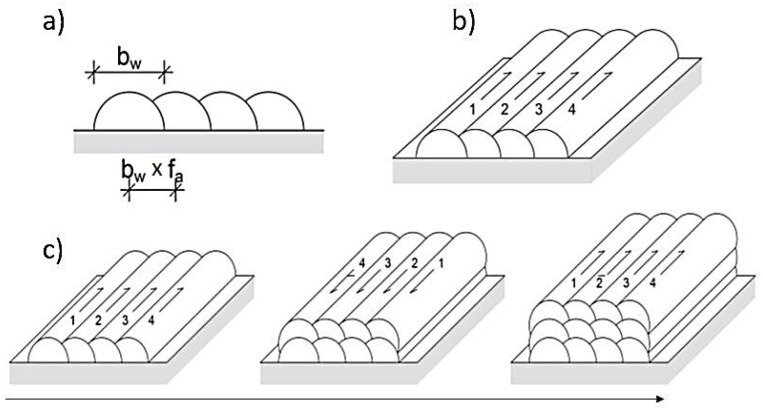
The schematic of the welding configuration: (**a**) weld bead and track spacing, (**b**) single-layer multi tracks, and (**c**) multi-layer multi tracks weld. The arrows show the sequence and the direction of the welding. Arrows 1–4 show the direction of welding.

**Figure 5 materials-16-04862-f005:**
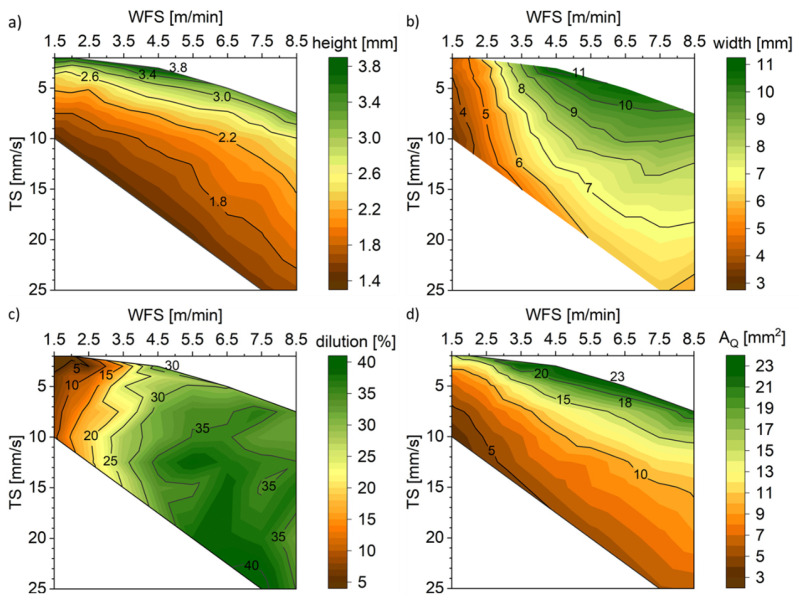
Measured height (**a**) and width (**b**) of the single-layer single-track weld seam and calculated values for dilution (**c**) and cross-sectional area A_Q_ (**d**).

**Figure 6 materials-16-04862-f006:**
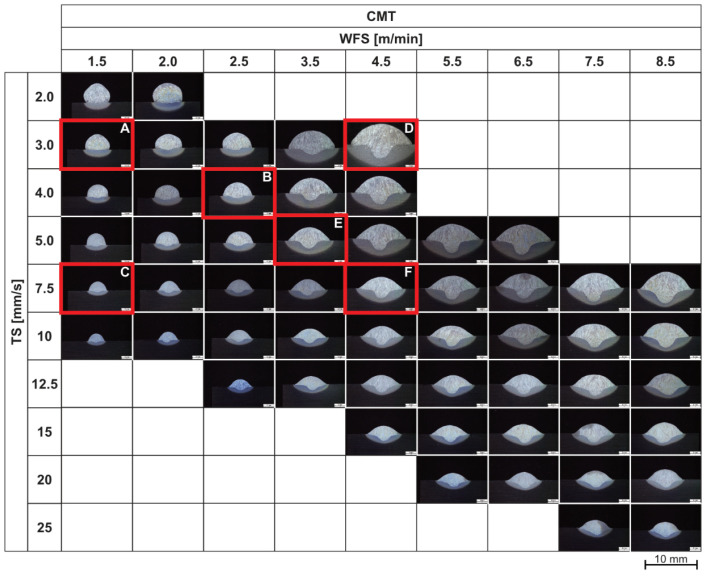
Cross-section of highlighted parameter (A–F) of CMT single-layer single-track welds and selected parameters for further investigations.

**Figure 7 materials-16-04862-f007:**
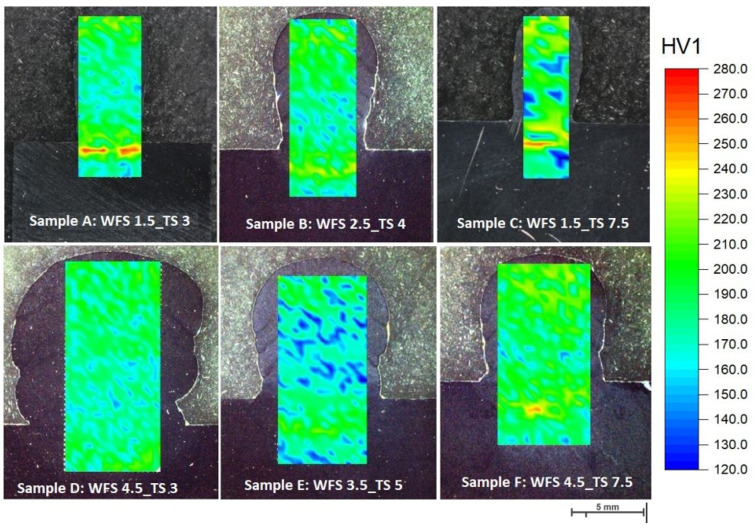
Overall view of hardness maps for all welding parameters.

**Figure 8 materials-16-04862-f008:**
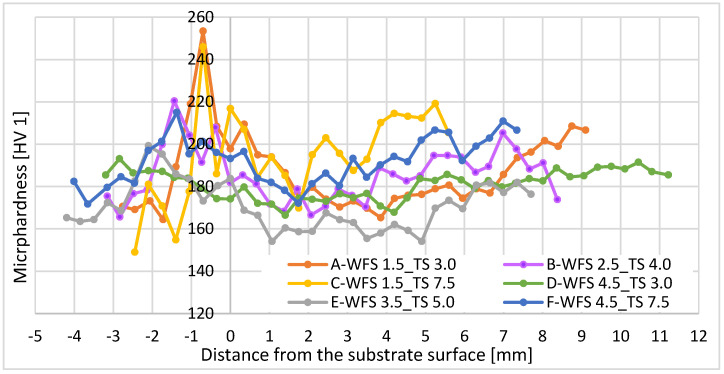
Microhardness versus distance from the substrate surface for all parameters A to F.

**Figure 9 materials-16-04862-f009:**
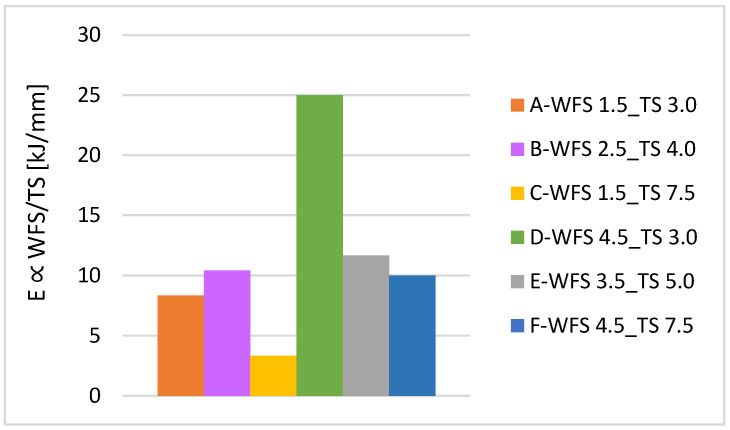
The energy input per unit length for all parameters.

**Figure 10 materials-16-04862-f010:**
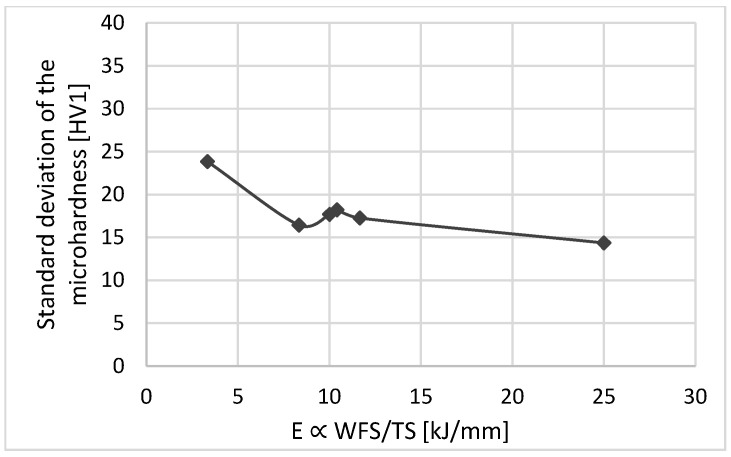
The relation between hardness distribution and energy input.

**Figure 11 materials-16-04862-f011:**
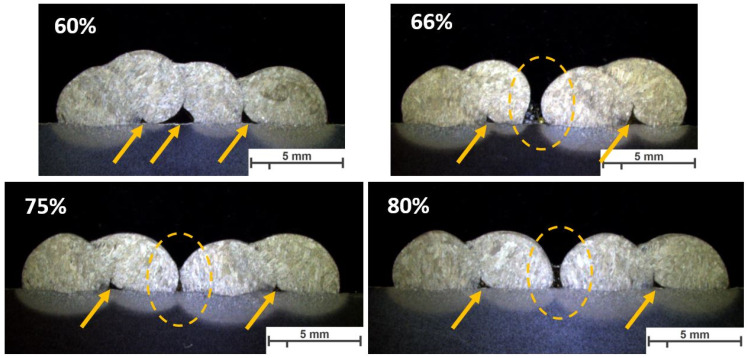
The macroscopic image of single layer-multi tracks welded with parameter A, WFS 1.5 m/min and TS 3.0 mm/s. Arrows and highlighted ovals show the lack of fusion regions.

**Figure 12 materials-16-04862-f012:**
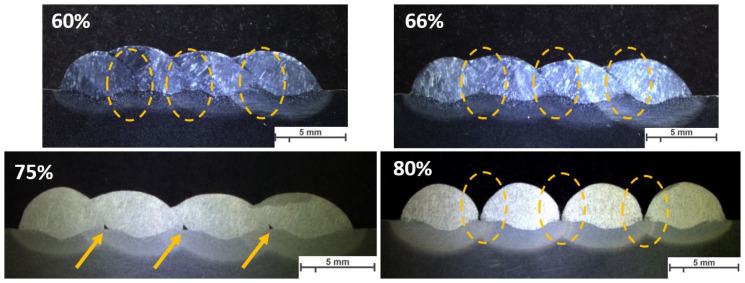
The macroscopic image of single layer-multi tracks welded with parameter B, WFS 2.5 m/min and TS 4.0 mm/s. Arrows and highlighted ovals show the lack of fusion regions.

**Figure 13 materials-16-04862-f013:**
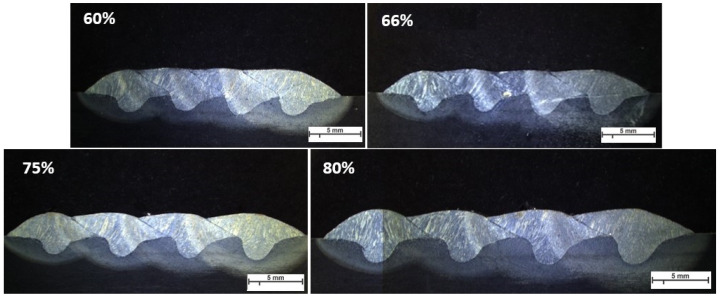
The macroscopic image of single layer-multi tracks welded with parameter F, WFS 4.5 m/min and TS 7.5 mm/s.

**Figure 14 materials-16-04862-f014:**
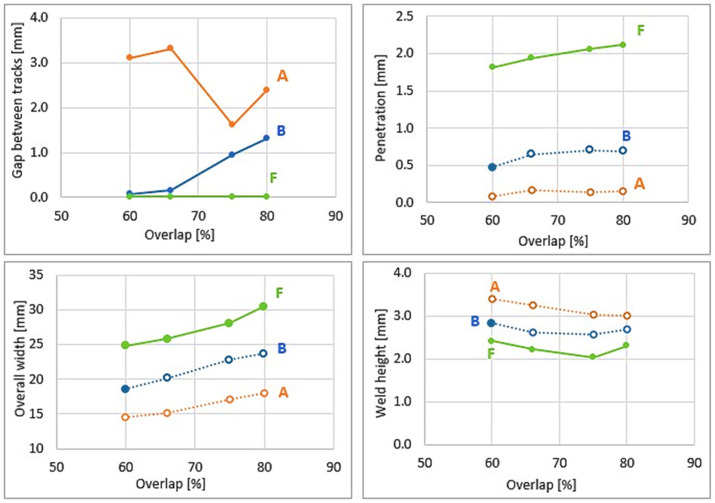
Geometry measurements of samples with the following: parameter A (orange): WFS 1.5 m/min–TS 3.0 mm/s; parameter B (blue): WFS 2.5 m/min–TS 4.0 mm/s; and parameter F (green): WFS 4.5 m/min–TS 7.5 mm/s.

**Figure 15 materials-16-04862-f015:**
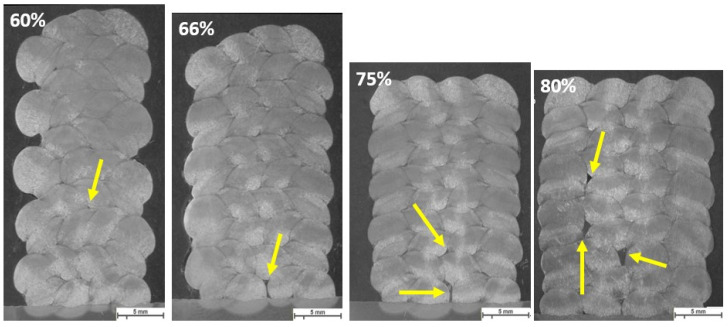
The macroscopic image of multi-layer multi-track welds with parameter A, WFS 1.5 m/min and TS 3.0 mm/s, and different overlaps (60%, 66%, 75%, 80%). Arrows show the lack of fusion regions. The scale bar represents 5 mm.

**Figure 16 materials-16-04862-f016:**
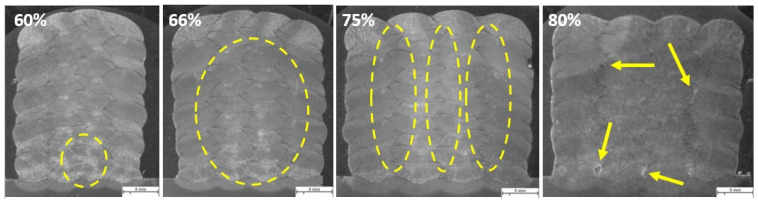
The macroscopic image of multi-layer multi-track welds with parameter B, WFS 2.5 m/min and TS 4.0, and different overlaps (60%, 66%, 75%, 80%). Arrows and highlighted ovals show the lack of fusion regions. The scale bar represents 5 mm.

**Figure 17 materials-16-04862-f017:**
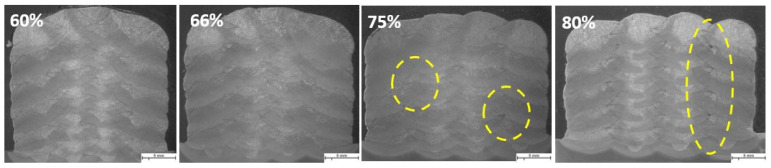
The macroscopic image of multi-layer multi-track welds with parameter F, WFS 4.5 m/min and TS 7.5 mm/s, and different overlaps (60%, 66%, 75%, 80%). Highlighted ovals show the lack of fusion regions. The scale bar represents 5 mm.

**Figure 18 materials-16-04862-f018:**
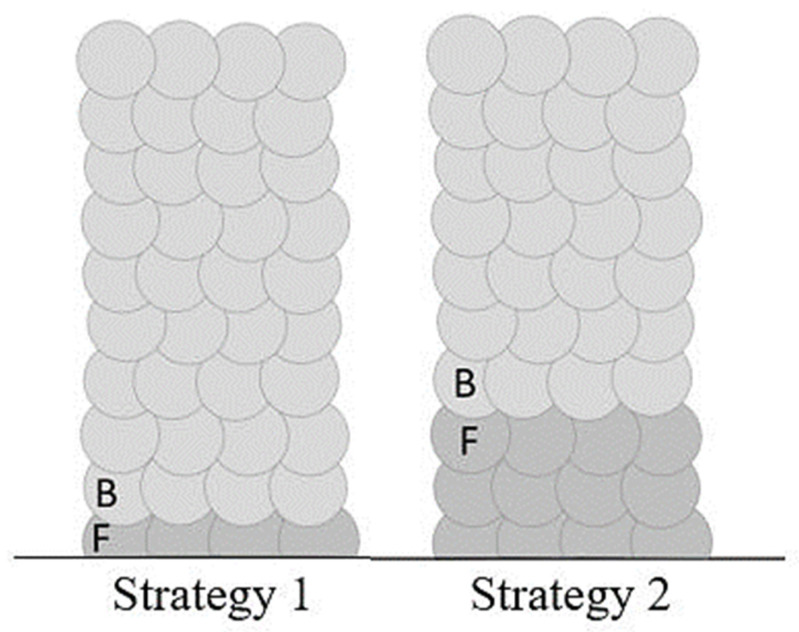
The schematic of welding with combined parameters.

**Figure 19 materials-16-04862-f019:**
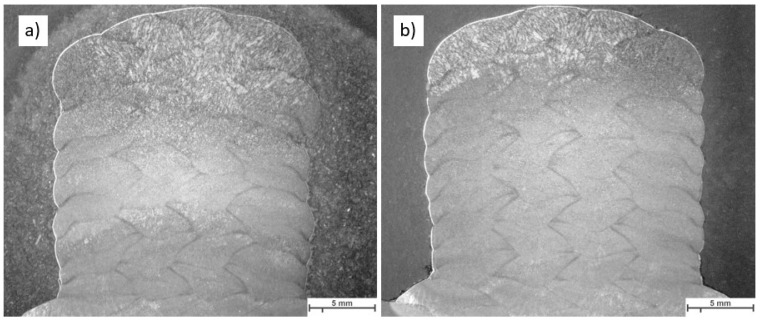
The macroscopic image of multi-layer and multi-track welds with parameter F for the first layer, and built up with parameter B (strategy 1); (**a**) 60% and (**b**) 66% overlap.

**Figure 20 materials-16-04862-f020:**
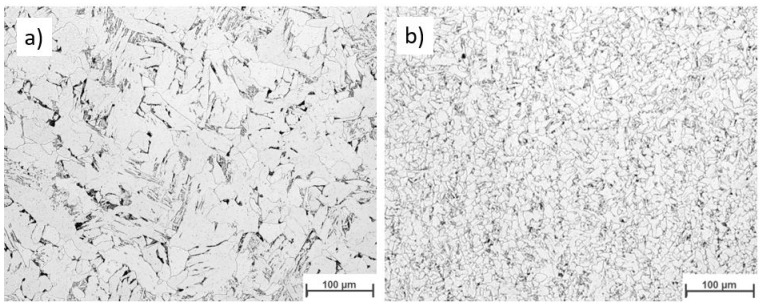
The microscopic images of the multi-layer multi-track welds according to strategy 1: 60% overlap, (**a**) top and (**b**) bottom.

**Figure 21 materials-16-04862-f021:**
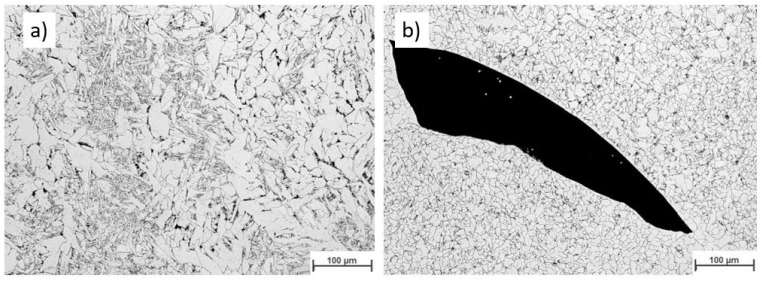
The microscopic images of the multi-layer multi-track welds according to strategy 1: 66% overlap, (**a**) top and (**b**) bottom.

**Figure 22 materials-16-04862-f022:**
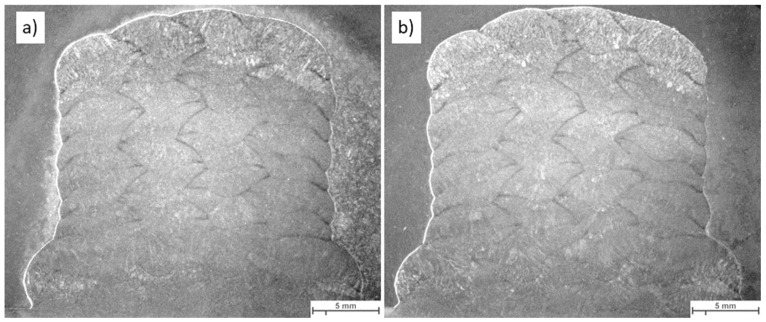
The macroscopic image of multi-layer multi-track welds with parameter F for the first three layers, and built up with parameter B (strategy 2): (**a**) 60% and (**b**) 66% overlap.

**Figure 23 materials-16-04862-f023:**
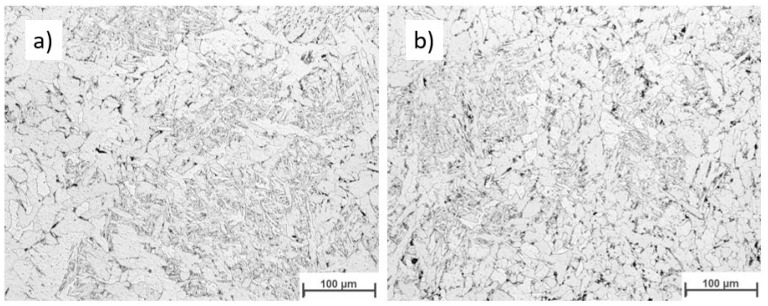
The microscopic image of the multi layers -multi tracks welded according to strategy 2: 60% Overlap, (**a**) top and (**b**) bottom.

**Figure 24 materials-16-04862-f024:**
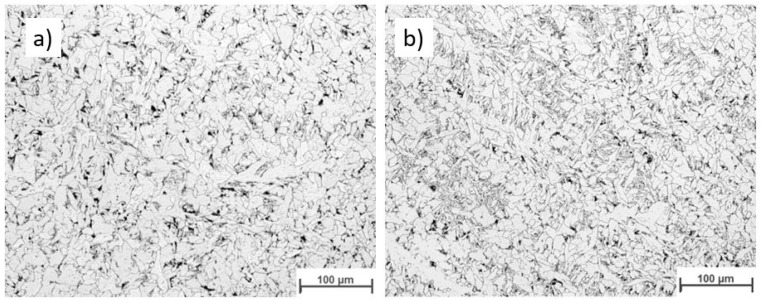
The microscopic image of the multi-layer multi-track welds according to strategy 2: 66% overlap, (**a**) top and (**b**) bottom.

**Table 1 materials-16-04862-t001:** Chemical composition of solid wire in wt%.

C	Mn	Si	P	S	Fe
0.08	0.9	1.45	0.013	0.0015	Bal.

**Table 2 materials-16-04862-t002:** Selected parameters based on wire feed speed (WFS) and travel speed (TS).

Sample	A	B	C	D	E	F
WFS (m/min)	1.5	2.5	1.5	4.5	3.5	4.5
TS (mm/s)	3.0	4.0	7.5	3.0	5.0	7.5

**Table 3 materials-16-04862-t003:** Current and voltage of different WFS during welding process.

	WFS (m/min)			
	1.5	2.5	3.5	4.5
Current (I)	60	96	131	161
Voltage (V)	10.9	12.1	13.2	13.9

## Data Availability

The data presented in this study are openly available in FigShare at https://figshare.com/s/26e82823ab68c9424c03 (accessed on 9 May 2023).

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
