# Peer review of "Selection of Parameters for Optimized WAAM Structures for Civil Engineering Applications"

_materials, 2023, doi:10.3390/ma16134862_

Round 1

Reviewer 1 Report

1.            In abstract, only qualitative results are stated. This section is modified by stating the quantitative results.

2.            Use more up-to-date references.

3.            Fig. 9-11, line no. 162. Authors have mentioned a lack of fusion was observed why it comes and why selected only parameter A, B and F. However authors discussed “higher the energy input per length, the more homogeneous the microhardness distribution is” in line no.154.

4.            Authors should maintain the gap between the words.  For Example: Line no. 199, Fig. 12, A(orange): WFS 1.5 m/min -TS3.0 mm/s., Line no.204. Figure 13Figure 15, Line No. 260; Figure 21Figure 22.

5.            What is the significance of the multilayer multi tracks found in macroscopic images?

6.            Author should include the equipment detail used for macroscopic analysis and scale should be visible also.

7.            Author should elaborate the microscopic images of Fig.18, 19, 21 and 22 in details.

Author Response

  1. In abstract, only qualitative results are stated. This section is modified by stating the quantitative results.

Answer: added to the abstract:

The microhardness of different welding parameters relies between 150 HV1 and 250 HV1. The optimum energy input per unit length is offered as the ratio between the wire feed speed (WFS) and the travel speed (TS) is suggested as 10. The influence of welding setup according to optimum energy input is discussed. The defects are analyzed with 60%, 66%, 75%, and 80% overlap between welding tracks. Welds integrity is investigated, and two strategies with combined welding parameters are determined. Welding defects in the preliminary layers with 66% overlap, while samples with 60% overlap re-vealed no defects. Strategy 2 did not contain defects throughout the whole cross-section area of the welds for both 60% and 66% overlaps.

  1. Use more up-to-date references.

Done!

  1. Fig. 9-11, line no. 162. Authors have mentioned a lack of fusion was observed why it comes and why selected only parameter A, B and F. However authors discussed “higher the energy input per length, the more homogeneous the microhardness distribution is” in line no.154.

Answer: Added to the section 4.2:

The higher the energy input per length, the more homogeneous the microhardness distribution is. However, the parameter with the homogenous hardness distribution, i.e., parameter D, consumes more material, and the weld build-up increases in width and height. On the other hand, although parameter C leads to the lowest energy input, it results in nonhomogenous hardness distributions. Parameter selection is conducted considering weld integrity and wettability, as well as the lowest possible energy input. Therefore, considering the optimum energy input per length, i.e., proportional to 10 kJ/mm, the experimental window was confined to parameters A, B, E, and F. Finally, parameter E was excluded from the selection due to its nonhomogenous hardness distribution.

  1. Authors should maintain the gap between the words.  For Example: Line no. 199, Fig. 12, A(orange): WFS 1.5 m/min -TS3.0 mm/s., Line no.204. Figure 13Figure 15, Line No. 260; Figure 21Figure 22.

Answer: It has been modified.

  1. What is the significance of the multilayer multi tracks found in macroscopic images?

Answer: Added to the section 4.5:

These results can help correlate the weld integrity concerning the welding parameters and structure. Although all the selected parameters have approximately identical energy input per unit length, they lead to different welding integrity. This knowledge could assist to build up a structure with efficient energy consumption and higher integrity.

  1. Author should include the equipment detail used for macroscopic analysis and the scale should be visible also.

Answer: equipment details are added in section 2. Scales are modified and also added to the image captions.

  1. Author should elaborate on the microscopic images of Fig.18, 19, 21, and 22 in details.

Answer: added in section 4.6.

The microstructure of the samples consists of ferrite grains (white regions) and pearlite islands (black regions) throughout the entire cross-section. In the lower part of the weld, ferrite grains are homogeneously equiaxed because of the tempering effect. In the upper region, acicular ferrite was observed due to faster cooling.

Thank you very much for the nice remarks.

Reviewer 2 Report

This paper conducts an experimental study on WAAM process, on the technological area more than on scientific one. The use of CMT mode for a GMAW process is, now, very classical for such an application. There is no novelty in this paper. A long time ago, maybe much more than 20 years, this kind of process was known as "reloading" or "repairing", or "cladding", but today, it is called as "additive manufacturing". In this work, only blocks are manufactured, we may wonder about the interest to do that and to put forward "additive manufacturing" in the title of this article . However, authors have to consider following points:

line 126; Which W to H ratio, integrity, flank angle is suitable for  WAAM process and why ? For example,  for condition WFS=4,5 m/min, TS= 3 and TS=7,5 m/min are suitable, but not TS =4 and 5 m/min (?), same question for WFS = 1,5 m/min.

line 148; please, give I and U , Frequency of CMT process parameter , or at least, average value. 

line 153-154; you write that homogeneity of hardness value distribution is linked to E, but you do not give standard deviation to illustrate that, at least. I think you do not have to place reference in this paragraph, references have to be placed in introduction part. 

Figure 7, only 1 point in HAZ, it is not sure you get max hardness value in this place.

Figure 8 presents no interest. Replace by a table.

& 4.3; I do not understand the interest to consider multitrack A and B. clearly, it was not necessary to make a  preparation metallographic. For example, figure 9, it is visible that A weld track is not good.

&4.4; idem

line 182->184; place references in introduction part.

&4.5; If tracks made with parameter A and B present defect, why to continue and to make blocks with A and B conditions? 

line 232, 60% and 66% overlap condition, is this enough significant variation to conclude about the choice of this parameter? which size does represent 6% overlap? 

&5; 

line 275; TFS impacts hardness distribution, according to authors, but they do not give any value illustrating this statement. They test 2 overlap conditions (60 and 66%) very close to each other and they make no replicates to estimate the significance. 

line 284, it is an evidence that different parameters have to be selected in WAAM building structure , due to the fact that first tracks are build on a cold bulk substrate, then, after several tracks, substrate + first layers change temperature.

There is only 2 references/8 about WAAM process ranging from 2013 to 2021. I get 1060 results about "WAAM" only in 2023 after a  scholar google search. First, the literature review has to be significantly improved before to consider a publication. Secondly,  there is no mechanical test of the builded WAAM structure (e.g., tensile test, at least) in this work , hardness measurement is not sufficient, there is not the possibility to estimate anisotropy of the mechanical properties for such obtained stratified materials.

Author Response

Reviewer 3

This paper conducts an experimental study on WAAM process, on the technological area more than on scientific one. The use of CMT mode for a GMAW process is, now, very classical for such an application. There is no novelty in this paper. A long time ago, maybe much more than 20 years, this kind of process was known as "reloading" or "repairing", or "cladding", but today, it is called as "additive manufacturing". In this work, only blocks are manufactured, we may wonder about the interest to do that and to put forward "additive manufacturing" in the title of this article . However, authors have to consider following points:

line 126; Which W to H ratio, integrity, flank angle is suitable for  WAAM process and why ? For example,  for condition WFS=4,5 m/min, TS= 3 and TS=7,5 m/min are suitable, but not TS =4 and 5 m/min (?), same question for WFS = 1,5 m/min.

This investigation was a part of another work that my colleague already published here:

Holzinger, C. et al., ‘3DWelding // Additve Fabrication of Structural Steel Elements in: ASMET [Hrsg.] Metal Additive Manufacturing Conference 2022. TU Graz. ASMET- Austrian Society for Metallurgy and Materials, (2022) S. 9–18.

line 148; please, give I and U , Frequency of CMT process parameter , or at least, average value. 

We introduced I and U to define the relation between P and WFS. These values were considered constant. The mean deposition current was 120 A, and the average deposition voltage was 17.9-18 V. The average CMT frequency was about 80 Hz. Since these parameters were not considered variables, we did not bring them to the discussion.

line 153-154; you write that homogeneity of hardness value distribution is linked to E, but you do not give standard deviation to illustrate that, at least. I think you do not have to place references in this paragraph, references have to be placed in the introduction part. 

Thank you for the remark. We believe it is permitted to use references in the results and discussion section. However, the linear relation between P and WFS is not an outcome of our results. Regarding the standard deviation to link hardness to the E, I have to admit each point on the hardness plot, is the average of at least 10 measurements along 1 line (of the same height/depth from the baseline). I added a plot (fig. 10) to illustrate the relationship. if I understood your comment correctly.

Figure 7, only 1 point in HAZ, it is not sure you get the max hardness value in this place.

Thank you for this comment, I have to point out that each point represents the average of 11 measurements along one line, which means the represented hardness is not a single value but an average. It might be attributed to the fact that the processes with lower heat input cool faster, leading to a smaller HAZ as well as higher hardness, whereas high heat input have a slower cooling rate, leading to a lower hardness (figure 7, 1 mm below the substrate surface).

Figure 8 presents no interest. Replace by a table.

I am sorry but I did not get it. Table of the energy in terms of the parameters? The energy is calculated with respect to the parameters (eq 3). The table will only repeat the calculations, while with the plot we found the visualization easier to being followed. However, the figure is one of the other reviewers’ interests and we have already applied their comments on it.

& 4.3; I do not understand the interest to consider multitrack A and B. Clearly, it was not necessary to make a  preparation metallographic. For example, figure 9, it is visible that A weld track is not good.

Thank you for mentioning that. The aim of this work was to systematically study the effect of the parameters on both single and multitrack in single and multi-layer build-up. We selected A, B, and F based on the first energy as well as wire consumption. Then we are aiming to discuss the results by characterizing them precisely. I agree it the parameter A it is obvious that even in single layer welds we do not observe weld integrity which is also an outcome of the work. Then we proceed with building up blocks since we observed the welds differs with different welding strategies, even though we keep the parameter constant (the same with B, figures 12 and 16).

& 4.4; line 182->184; place references in introduction part.

The reference is cited in the result and discussion section. We did not discuss the E in the introduction.

&4.5; If tracks made with parameters A and B present defect, why to continue and to make blocks with A and B conditions? 

As we mentioned in & 4.3; we admit there are clear weld defects in the single layer. However, the weld results in different characteristics between single-layer and multilayer build due to the tempering effect of the posterior layers, the speed of the building up, the direction of the welding, and so on (as we also see the difference between the type of the weld integrity in a single layer and multi-layer of parameter B). For instance, in this case, the shape, and the size of the defects alternate only by shifting from a single to a multi-layer strategy (figures 12 and 16 for example overlap 80%). The aim was to entirely investigate and discuss all the results.

line 232, 60% and 66% overlap condition, is this enough significant variation to conclude about the choice of this parameter? which size does represent 6% overlap? 

We selected the parameters in terms of the energy and wire consumption (weld shape and geometry), over the studied overlaps and based on their better results 2 out of the 4 overlaps are selected.

&5; 

line 275; TFS impacts hardness distribution, according to authors, but they do not give any value illustrating this statement. They test 2 overlap conditions (60 and 66%) very close to each other and they make no replicates to estimate the significance. 

It has been added in section 4.2 lines no. 182 to 194. According to the abovementioned answer of the reviewer: comment line 153-154;

line 284, it is an evidence that different parameters have to be selected in WAAM building structure , due to the fact that first tracks are build on a cold bulk substrate, then, after several tracks, substrate + first layers change temperature.

There is only 2 references/8 about WAAM process ranging from 2013 to 2021. I get 1060 results about "WAAM" only in 2023 after a scholar google search. First, the literature review has to be significantly improved before to consider a publication.

Literature review has been modified.

Secondly,  there is no mechanical test of the builded WAAM structure (e.g., tensile test, at least) in this work , hardness measurement is not sufficient, there is not the possibility to estimate anisotropy of the mechanical properties for such obtained stratified materials.

The mechanical tests are the outlook of this work and in progress to be published. We have mentioned in the state of the art that this work forms a basis for the adaptation of the welding parameters to the mechanical properties of the materials and serves as input knowledge for further investigations.

Thank you very much for the nice remarks.

Reviewer 3 Report

This work related to research the influence of the processing parameters of the WAAM technique on the microhardness and geometry of the weld.

In general, the article deserves a good rating, especially in terms of results and discussion.

However, it must be subjected to major revision

The reviewer asks to pay attention to the following points

Introduction needs to be expanded

It is necessary to describe the process (for example - https://doi.org/10.3390/ma12071121)

Attention should be paid to ways to control properties after printing using annealing (for example - https://doi.org/10.1007/s40830-022-00363-4)

It is necessary to write in more detail the motivation and novelty of the study - how does your article differ from those already published - it is necessary to perform a detailed analysis https://doi.org/10.1016/j.istruc.2022.08.084 - please review at least 5-7 sources

Goal needs to be more clear

Research methods and equipment are indicated in section 3 - should be in section 2

In figure 6 and 8 - need to add units of measurement

It is also necessary to specify the hardness of the original wire - 

What is the reason for the sharp increase in hardness in figure 7 at a distance of -1 mm?

In conclusion, it is necessary to indicate the speed values and explain what strategy 1 and 2 mean

Author Response

  1. Introduction needs to be expanded

Done!

  1. It is necessary to describe the process (for example - https://doi.org/10.3390/ma12071121)

Answer: Added in lines no 42-49: Wire-arc additive manufacturing (WAAM) is an efficient method for engineering structure production. The process can produce near-net-shaped components without complex tools, with lower costs and time consumption. In this method, a robotic arm controls the process, and the shape is built upon a substrate material. WAAM is the process of depositing layers by melting metal wire using an electric arc as the heat source to 3D print metal components. Figure illustrates a schematic of WAAM process.

  1. Attention should be paid to ways to control properties after printing using annealing (for example - https://doi.org/10.1007/s40830-022-00363-4)

Thank you for the reference and the comment. In the current study, we tried to design strategy to manufacture components that will not need post-processing treatment. In this concept, the method and welding configuration we designed enables us to either build small features on large components or the whole an engineering structure which both are needless to the post treatment. However, considering the material (steel), this could be included in another study scope.

  1. It is necessary to write in more detail the motivation and novelty of the study - how does your article differ from those already published - it is necessary to perform a detailed analysis https://doi.org/10.1016/j.istruc.2022.08.084 - please review at least 5-7 sources

Done!

  1. Goal needs to be more clear

Answer: Added in lines no 50-57

  1. Research methods and equipment are indicated in section 3 - should be in section 2

Answer: done!

  1. In figure 6 and 8 - need to add units of measurement

Answer: added

  1. It is also necessary to specify the hardness of the original wire – 

Answer: Thank you very much for your comment. Regarding the hardness of the original wire, the wire is produced by rolling and drawing process, and the melting during WAAM process resets all the mechanical properties which were resulted from previous cold deformation. Therefore, the only characterization which remains the same is the chemical composition, and the wire property will be independent of the original state, and related to the cooling rate during the process.

  1. What is the reason for the sharp increase in hardness in figure 7 at a distance of -1 mm?

Answer: added to line no. 171

The hardness in HAZ increases in samples welded with parameter A and C due to the lower heat input. The processes with lower heat input cool faster, leading to a higher hardness, whereas high heat input have a slower cooling rate, leading to a lower hardness (figure 7, 1 mm below the substrate surface).

  1. In conclusion, it is necessary to indicate the speed values and explain what strategy 1 and 2 mean

Answer: Added to lines no. 326-335

Experiments also showed that process parameters, e.g., travel speed, wire feed speed, and subsequently the energy input per unit length, influence the bead area in the WAAM process. Two strategies were selected to optimize welding energy as well as wire consumption and to acquire an intact weld (free of lack of fusion regions). The microscopic investigations of welding according to strategy 1 demonstrated defects in the preliminary layers with 66% overlap, while samples with 60% overlap revealed no defects. Strategy 2 did not contain defects throughout the whole cross-section area of the welds for both 60% and 66% overlaps. Finally, welding the first three layers with WFS of 4.5 m/min and TS of 7.5 mm/s, and continuing with WFS of 2.5 m/min and TS of 4.0 mm/s was selected as the optimized welding parameters configuration.

Thank you so much for the appropriate remarks.

Reviewer 4 Report

The authors have made a good attempt at selection of parameters for WAAM structures but the paper in it's current version needs major revisions. Some comments below: 
  1. Lines 148 - 152: The authors need to do a better job articulating the terms in sentences and linking them to equations. 
  2. Some figures lack units (example: Figure 8, y axis)
  3. The scale bars are a bit small - please make them bigger in all images 
  4. The authors overall need to dig deeper into discussion sections and connect them prior studies (the unsually low number of references is a bit concerning - while not a major issue, it would be good to connect to other studies out there or explain to us why this is not feasible, if so)
  5. Figure 6, what's the unit of hardness, what are the depths for hardness and can the authors confirm that the plastic zone captured all microstructures of interest? 
  6. I find the overall link b/w structure and property to be a bit weak. The authors will need further bolster their discussion sections. 
  7. Can the authors extract any more quantitative information from the optical microscopy images. 

None 

Author Response

The authors have made a good attempt at selection of parameters for WAAM structures but the paper in it's current version needs major revisions. Some comments below: 

  1. Lines 148 - 152: The authors need to do a better job articulating the terms in sentences and linking them to equations. 

Answer: I expanded the explanation (line no. 183-199)

  1. Some figures lack units (example: Figure 8, y axis)

Answer: The units have been modified.

  1. The scale bars are a bit small - please make them bigger in all images 

Answer: I modified them. Additionally, I also added ‘The scale bar represents 5mm' in the caption

  1. The authors overall need to dig deeper into discussion sections and connect them prior studies (the unsually low number of references is a bit concerning - while not a major issue, it would be good to connect to other studies out there or explain to us why this is not feasible, if so)

Answer: Thank you very much for this remark, I have improved the introduction and state of the art with respect to others' work and more references.

  1. Figure 6, what's the unit of hardness, what are the depths for hardness and can the authors confirm that the plastic zone captured all microstructures of interest? 

Answer: The units have been modified. The Plastic zone differs for different WFS and TS and so does the hardness depth. I measured the microhardness of each sample at least up to 5 rows beneath the plastic zone boundary (the distance between the rows is 0.35 µm). I will point it out in the methods (lines no. 107-109). Thank you for mentioning that.

  1. I find the overall link b/w structure and property to be a bit weak. The authors will need further bolster their discussion sections. 

Answer: We discussed the microstructure and the effect of tempering of the layers in the discussion of combined layers (line no. 310-315 and in the conclusion too)

  1. Can the authors extract any more quantitative information from the optical microscopy images. 

Answer: I have measured the size of the defects, and have added it to line no. 316

Thank you very much for your corrections.

Reviewer 5 Report

The work presents improvements and could be accepted in the current format. 

Author Response

Thank you very much for your positive feedback.

Round 2

Reviewer 2 Report

Sorry, I did not detect a substantial improvement  in your article. 

Author Response

Round 2:

Sorry, I did not detect a substantial improvement  in your article. 

Answer:

Dear reviewer, Thank you for your feedback. We have addressed all remarks in the first round’s response. Please specify any remaining unmodified points for us to reform. The current version received approval from four other reviewers; therefore, we kindly ask you to provide the remarks to help us finalize the paper.

Best

**************************************************************

Round 1:

This paper conducts an experimental study on WAAM process, on the technological area more than on scientific one. The use of CMT mode for a GMAW process is, now, very classical for such an application. There is no novelty in this paper. A long time ago, maybe much more than 20 years, this kind of process was known as "reloading" or "repairing", or "cladding", but today, it is called as "additive manufacturing". In this work, only blocks are manufactured, we may wonder about the interest to do that and to put forward "additive manufacturing" in the title of this article . However, authors have to consider following points:

  1. line 126; Which W to H ratio, integrity, flank angle is suitable for  WAAM process and why ? For example,  for condition WFS=4,5 m/min, TS= 3 and TS=7,5 m/min are suitable, but not TS =4 and 5 m/min (?), same question for WFS = 1,5 m/min.

Answer: This investigation was a part of another work that my colleague already published here:

Holzinger, C. et al., ‘3DWelding // Additve Fabrication of Structural Steel Elements in: ASMET [Hrsg.] Metal Additive Manufacturing Conference 2022. TU Graz. ASMET- Austrian Society for Metallurgy and Materials, (2022) S. 9–18.

  1. line 148; please, give I and U , Frequency of CMT process parameter , or at least, average value. 

Answer: Added in table 3.

  1. line 153-154; you write that homogeneity of hardness value distribution is linked to E, but you do not give standard deviation to illustrate that, at least. I think you do not have to place reference in this paragraph, references have to be placed in introduction part. 

Answer: Thank you for the remark. However, the linear relation between P and WFS is not an outcome of our results. Regarding the standard deviation to link hardness to the E, I have to admit each point on the hardness plot, is the average of at least 10 measurements along 1 line (of the same height/depth from the base line). I added a plot (fig. 10) to illustrate the relationship. if I understood your comment correctly.

Regarding the reference, we see it is permitted to use references in the results and discussion section.

  1. Figure 7, only 1 point in HAZ, it is not sure you get max hardness value in this place.

Answer: Thank you for this comment, I have to point out each point represent the average of 10 measurements along one line, which means the represented hardness is not a single value but an average. It might be attributed to the fact that the processes with lower heat input cools faster, leading to a smaller HAZ as well as higher hardness, whereas high heat input have a slower cooling rate, leading to a lower hardness (figure 7, 1 mm below the substrate surface).

  1. Figure 8 presents no interest. Replace by a table.

Answer: I am sorry but I did not get it. Table of the energy in terms of the parameters? The energy is calculated with respect to the parameters (eq 3). Table will only repeat the calculations, while with the plot we found the visualizing easier to being followed. In addition, the figure is one of the other both reviewers’ interest and we have already applied their comments on it.

  1. & 4.3; I do not understand the interest to consider multitrack A and B. clearly, it was not necessary to make a  preparation metallographic. For example, figure 9, it is visible that A weld track is not good.

Answer: Thank you for mentioning that. The aim of this work was to systematically study the effect of the parameters on both singe and multitrack in single and multi-layer build up. We selected A, B, and F based on the first energy as well as wire consumption. Then we are aiming to discuss the results by characterizing them precisely. I agree it the parameter A it is obvious that even in single layer welds we do not observe weld integrity which is also an outcome of the work. Then we proceed with building up blocks since we observed the welds differs with different welding strategies, even though we keep the parameter constant (the same with B, figures 12 and 16).

  1. & 4.4; line 182->184; place references in introduction part.

Answer: The reference is cited in result and discussion section. We did not discuss the E in the introduction.

  1. &4.5; If tracks made with parameter A and B present defect, why to continue and to make blocks with A and B conditions? 

Answer: As we mentioned in & 4.3; we admit there are clear weld defects in the single layer. However, the weld results different characteristics between single layer and multilayer builds up due to the tempering effect of the posterior layers, the speed of the building up, the direction of the welding and so on (as we also see the difference between the type of the weld integrity in single layer and multi-layer of parameter B). For instance, in this case, the shape, and the size of the defects alternates only by shifting from single to multi layer strategy (figures 12 and 16 for example overlap 80%). The aim was to entirely investigate and discuss all the results.

  1. line 232, 60% and 66% overlap condition, is this enough significant variation to conclude about the choice of this parameter? which size does represent 6% overlap? 

Answer: We selected the parameters in terms of the energy and wire consumption (weld shape and geometry), over the studied overlaps and based on their better results 2 out of the 4 overlaps are selected.

  1. line 275; TFS impacts hardness distribution, according to authors, but they do not give any value illustrating this statement. They test 2 overlap conditions (60 and 66%) very close to each other and they make no replicates to estimate the significance. 

Answer: It has been added in section 4.2 lines no. 182 to 194. According to the abovementioned answer of the reviewer: comment line 153-154;

  1. line 284, it is an evidence that different parameters have to be selected in WAAM building structure , due to the fact that first tracks are build on a cold bulk substrate, then, after several tracks, substrate + first layers change temperature.

Answer: Yes. Exactly, we agree. In the current work, we aimed to investigate the influence of parameter configuration on the weld integrity and energy efficiency. This is an outcome of the work.

  1. There is only 2 references/8 about WAAM process ranging from 2013 to 2021. I get 1060 results about "WAAM" only in 2023 after a scholar google search. First, the literature review has to be significantly improved before to consider a publication.

Answer: Literature review has been modified.

  1. Secondly,  there is no mechanical test of the builded WAAM structure (e.g., tensile test, at least) in this work , hardness measurement is not sufficient, there is not the possibility to estimate anisotropy of the mechanical properties for such obtained stratified materials.

Answer: The mechanical tests are the outlook of this work and in progress to be published. We have mentioned in the state of the art that this work forms a basis for the adaptation of the welding parameters to the mechanical properties of the materials and serves as input knowledge for further investigations.

Thank you very much for the nice remarks.

Reviewer 3 Report

The article has been improved, the authors took into account many comments

graphs are often hard to perceive, there are no error bars

Author Response

The article has been improved, the authors took into account many comments

  1. graphs are often hard to perceive, there are no error bars.

Answer: Thank you very much for your positive feedback. I have added a graph of standard deviation for microhardness measurements (figure 10). For the rest, the geometry measurements were single measurement of the width, depth, height and the gap between tracks.

Reviewer 4 Report

No more edits.